# What Is in a Cat Scratch? Growth of *Bartonella henselae* in a Biofilm

**DOI:** 10.3390/microorganisms9040835

**Published:** 2021-04-14

**Authors:** Udoka Okaro, Sierra George, Burt Anderson

**Affiliations:** 1Foundational Sciences Directorate, Bacteriology Division, United States Army Medical Research Institute of Infectious Diseases, Frederick, MD 21702, USA; udokaokaro@gmail.com; 2Department of Molecular Medicine, MDC7, Morsani College of Medicine, University of South Florida, 12901 Bruce B. Downs Blvd., Tampa, FL 33612, USA; sierrageorge@usf.edu

**Keywords:** *Bartonella henselae*, cat flea, sRNA, biofilm formation, gene regulation, transcription terminator, transcription factor, trimeric auto transporter adhesin

## Abstract

*Bartonella henselae* (*B. henselae*) is a gram-negative bacterium that causes cat scratch disease, bacteremia, and endocarditis, as well as other clinical presentations. *B. henselae* has been shown to form a biofilm in vitro that likely plays a role in the establishment and persistence of the bacterium in the host. Biofilms are also known to form in the cat flea vector; hence, the ability of this bacterium to form a biofilm has broad biological significance. The release of *B. henselae* from a biofilm niche appears to be important in disease persistence and relapse in the vertebrate host but also in transmission by the cat flea vector. It has been shown that the BadA adhesin of *B. henselae* is critical for adherence and biofilm formation. Thus, the upregulation of *badA* is important in initiating biofilm formation, and down-regulation is important in the release of the bacterium from the biofilm. We summarize the current knowledge of biofilm formation in *Bartonella* species and the role of BadA in biofilm formation. We discuss the evidence that defines possible mechanisms for the regulation of the genes required for biofilm formation. We further describe the regulation of those genes in the conditions that mimic both the arthropod vector and the mammalian host for *B. henselae*. The treatment for persistent *B. henselae* infection remains a challenge; hence, a better understanding of the mechanisms by which this bacterium persists in its host is critical to inform future efforts to develop drugs to treat such infections.

## 1. Introduction

Since the first report of microbial biofilms nearly 40 years ago, two decades passed before interest began to grow in studies that examined the clinical significance of biofilm formation [1]. Studies that elucidate the complexity and dynamics of bacterial biofilms have continued to grow in recent years. As a result, increased data has become available establishing the intricate relationship between gene regulation, biofilm formation, and disease progression.

The genus *Bartonella* consists of numerous species, some of which are known to cause Trench fever, Carrion’s disease, and cat scratch disease (CSD) [2]. Trench fever, originally described more than 100 years ago as infecting nearly one million troops during World War I, is caused by *B. quintana* [3]. Evidence of Carrion’s disease can be traced back to pre-Inca cultures, but the illness was not attributed to infection with *B. bacilliformis* until the early 1900s [4]. CSD caused by *B. henselae* remains one of the most common infections caused by bacteria in the genus *Bartonella*. The Centers for Disease Control and Prevention (CDC) estimates more than 12,500 diagnosed cases of CSD annually in the US, although the disease is prevalent worldwide [5,6,7,8]. Recently, *Bartonella* species have been isolated from a wide array of species ranging from terrestrial animals to sea inhabitants, demonstrating the ability of *Bartonella* to adapt and survive in a diverse range of hosts [9,10,11].

Amongst the almost 50 defined or proposed *Bartonella* species, *B. henselae* is a gram-negative, intracellular zoonotic bacterium that infects both cats and humans [12]. Cats, the natural reservoir of *B. henselae*, generally do not show symptoms of infection. In several case reports, however, *B. henselae* has been isolated in cats with a variety of clinical symptoms including endocarditis, seizure disorders, ocular disease, and hyperglobulinaemia [13]. *B. henselae* infection occurs in humans when the bacterium is accidently transmitted through the scratch or bite of a cat, but other modes of transmission such as red ant bites have been reported [14]. Flea infestation on domestic animal reservoirs such as cats and dogs are of rising concern, as resistance to flea control insecticides is increasing [15].

*B. henselae* is able to adhere to host cells and form a biofilm (Figure 1) [16,17,18]. The ability of *B. henselae* or any microbe to form a biofilm has been linked to chronic diseases [19,20]. Upon diagnosis with a systemic *B. henselae* infection such as endocarditis, patients may be treated with ciprofloxacin or azithromycin depending on the severity of symptoms [21]. Treatment failure in patients diagnosed with *B. henselae* endocarditis has been largely attributed to the ability of *B. henselae* to form a biofilm and resist antibiotics. Recent in vitro data suggests drug combination treatments are more effective in eliminating *B. henselae* chronic infection, compared to current single drug treatments [22]. Despite the continued use of antibiotics, systemic cases of *B. henselae* infection remain difficult to treat and often require more invasive treatment courses [23]. Biofilm-associated antibiotic resistance or tolerance is a major mechanism used by pathogenic bacteria located within the extracellular matrix or extracellular polymeric substance (EPS). This EPS matrix has been shown to increase antibiotic resistance or tolerance by several mechanisms including inhibiting antibiotic penetration, secreting enzymes to degrade antibiotics, and activating signaling pathways for adaptation and survival [24]. Biofilm regulation in *B. henselae* consists of a complex gene regulatory system that allows the bacteria to survive in different phases of the lifecycle, resist antibiotic treatment, and persist in the human host [19,25]. The ineffectiveness of current antibiotic treatment for biofilm associated infections warrants further investigation of the mechanism by which *B. henselae* regulates biofilm-associated genes.

Given the history of the *Bartonella* genus, the increased prevalence of *Bartonella* in a variety of species, and the poor treatment efficacy of systemic *B. henselae* infection, it is necessary to determine how *B. henselae* survives and persists, specifically through the formation of biofilms. Understanding such mechanism(s) will aid in the development of more effective treatments for *B. henselae* infection. Therefore, this review focused on the clinical significance, the prevalence of biofilms during the lifecycle, and the currently known genes important for biofilm regulation of *B. henselae*.

## 2. Clinical Importance

Three different *Bartonella* species—*B. bacilliformis*, *B. quintana*, and *B. henselae*—are the species most commonly associated with acute or chronic infections in humans [26]. In general, the severity of clinical outcomes correlates with the patient’s immune status; therefore, the more severe cases typically occur in immunocompromised individuals [27]. *B. henselae* is known to show several clinical manifestations, such as cat scratch disease (CSD), a condition characterized by lymphadenopathy and mostly reported in children [28,29], chronic lymphadenopathy [28,30], fever with persistent bacteremia [31], bacillary angiomatosis [32], neurological conditions [33], peliosis hepatitis [34], and life-threatening infective endocarditis, which is usually reported as blood-culture-negative endocarditis [31,35,36].

Fleas are competent vectors for numerous microbial pathogens. *Ctenocephalides felis* (*C. felis*), known as cat fleas, are an opportunistic blood feeder and the arthropod vector for *B. henselae;* however, other vectors such as ticks have been proposed [37,38,39,40,41,42]. Cats are reported to be the predominant host for *B. henselae*, but *B. henselae* has been isolated from a variety of hosts [43,44,45,46]. Most cats are asymptomatic when infected, but the exceptions usually develop a fever and local inflammation at the site of inoculum [46,47,48].

*Bartonella* infections begin with the inoculation of the bacteria, which is usually associated with the feeding of the arthropod vector on an infected cat. Bacterial load from the flea gut is excreted in the flea fecal matter and subsequently onto the cat [49]. Human infections occur through a cat scratch inoculating the host with bacteria from the flea fecal matter.

In the life cycle of *B. henselae*, biofilms are implicated in both the flea and the mammalian hosts. A previous report from our laboratory showed scanning electron micrographs with bacteria in the gut and bacterial biofilm in the fecal matter of laboratory i-fected cat fleas [19]. The ability of *Bartonella* to form biofilms in vertebrate hosts has been reported in the literature [50,51,52]. A mouse model of *B. tayolrii* infection was shown to demonstrate persistent bacteremia, liver lesions, and eventually death [51]. The masses observed in the liver and kidney appeared embedded in an amorphous matrix, which can be defined as a biofilm, providing an experimental model to study human disease progression in an immunocompromised host. In Edouard et al. (2015) [50], a study with 106 patients provided evidence of endocarditis by *B. quintana* and *B. henselae*. The above studies provide evidence that *Bartonella* biofilm communities are indeed an integral part of the vegetative mass associated with infective endocarditis [50,51,52]. *B. henselae* is a fastidious bacterium with particular nutritional requirements; hence, it is challenging to isolate and culture the bacterium from clinical samples [53]. Laboratory diagnosis is usually accomplished by one or more of the following diagnostic techniques: PCR, serology, isolation with extended incubation periods, or histopathology [54,55,56,57].

*Bartonella* species have a noteworthy ability to evade the host immune system and resist antimicrobial agents. While some isolates are susceptible to minocycline and macrolide antibiotics such as erythromycin, clarithromycin, azithromycin, and fluoroquinolone compounds with relatively low minimal inhibitory concentrations (MICs) [58,59,60], clinical experience has shown that *Bartonella* infection treatment failures are a major concern, as most of these antimicrobial classes exhibit only bacteriostatic properties [60,61,62]. It is expected that growth in biofilms allows *Bartonella* species to persist in the face of stress including antimicrobial treatment and the host immune response. Multiple studies and guidelines support 2 to 6 weeks of treatment for endocarditis using at least two antibiotics, one of which is an aminoglycoside [58,63,64,65]. Considering the clinical manifestations and antibiotic resistance of *B. henselae*, understanding the mechanisms by which *B. henselae* initiates biofilm formation is critical to comprehend how it causes chronic disease in humans.

## 3. Biofilm Formation, Composition, and Life Cycle

Bacteria grow as free-floating planktonic cells or as coordinated aggregates embedded in a matrix, referred to as a biofilm [66]. Biofilms can be single or multispecies communities, can thrive on most surfaces, and may be surface associated (agar, contact lens) or submerged under a static or shear flow condition (such as those formed in an artificial cardiac valve or indwelling catheter) [67,68]. Because of the diversity of biofilm surfaces, the well-organized structures of the colonies, and a characteristic anti-microbial resistance or tolerance, biofilms have received significant attention and are currently investigated for their role in infectious diseases [20,69]. Adhesion or aggregation is the first step of biofilm formation, virulence, and host cell interactions in most bacteria [70]. Outer membrane adhesins facilitate bacterial adhesion to a biotic or abiotic substrate and self-aggregation. Bacteria first adhere, then aggregate to allow the chemical signaling and quorum sensing communication required for the aggregates to secrete the proteins, polysaccharides, and extracellular DNA (eDNA) required for the assembly of the biofilm [71]. The *B. henselae* biofilm has been shown to contain proteins, polysaccharides, and eDNA, and both DNase and proteinase K have been shown to result in biofilm destruction [16].

The growth of *B. henselae* as a biofilm was first reported by Kyme et al. [72] as an auto-aggregative phase variation later linked to the expression of a surface adhesin called BadA. In the *Bartonella* genus, the roles of BadA and Vomps proteins are well documented for the adhesion steps in *B. henselae* and *B. quintana*, respectively. BadA is a trimeric auto transporter (TAA), and the role of TAAs have been significantly documented in other gram-negative bacteria [73,74,75]. A genetic deletion of *badA* in different strains of *B. henselae* led to failure to adhere efficiently and form a biofilm [12,16,76].

In humans, it has been speculated that *B. henselae* infects erythrocytes and may persist in these cells [77]. Evidence also supports persistence in endothelial progenitor cells, which presents the possibility for host immune system evasion [78,79,80]. *B. henselae* is speculated to form biofilms in the gut of the cat flea, which helps the bacteria persist and replicate in the gut [81]. The bacterial load is excreted in the fecal matter, where it forms a biofilm that protects the bacteria, which persists about 10–12 days in the flea fecal matter before human inoculation through the cat claw [19,49]. We propose that the biofilm represents an additional niche that provides the platform for seeding planktonic cells into the bloodstream, causing host immune reactions, disease conditions, and persistence in the face of antimicrobial treatments.

## 4. Genes Involved in Biofilm Regulation

Previously published data suggested that biofilm formation is regulated by cyclic diguanosine-5′-monophosphate (c-di-GMP), small RNAs (sRNAs), and quorum sensing (QS) in most bacterial species [82,83,84]. The role of both c-di-GMP and QS in biofilm formation has been adequately described in other gram-negative bacteria [85,86,87] but not in *B. henselae.* sRNAs have been associated with biofilm cellular regulation, such as general stress response and virulence [88]. In *E. coli.*, hundreds of sRNAs have been documented [89,90]. sRNAs work by primarily binding specific targets such as protein, RNA, or DNA, and a large portion of them require the Hfq protein to stabilize and facilitate interaction with their target [91]. sRNAs regulate biofilm formation in *S. typhimurium* and *E. coli* [92,93,94]. A high percentage of sRNA molecules within the bacterial genome are transcribed as non-coding small RNA (ncRNA) [94]. ncRNA may function as trans-acting antisense transcripts (asRNA) or as cis-acting RNA and the expression changes in response to environmental conditions such as pH, nutrient availability, antimicrobial peptides, and competition with other microbes, as reviewed in Ortega et al. (2014) [95].

In *Bartonella*, sRNAs have been acknowledged for playing a role in regulating genes required for bacterial transmission between vector and host [19,96] and biofilm formation [19,76]. A multicopy family of nine unannotated, well conserved, highly transcribed sRNAs termed Bartonella regulatory transcripts, Brts1-9, are implicated in *B. henselae* biofilm regulation. Using bioinformatics tools, all nine sRNAs were predicted to form a highly stable stem and loop structure at the end of the RNA, which is a characteristic feature of riboswitches [76]. The Brt RNAs were the fourth highest transcribed RNAs, which raises the question of why these multicopy sRNAs are transcribed at such levels. About 15 nucleotides downstream of each *brt* gene is a gene that codes for DNA-binding proteins named transcription regulatory proteins (*trps* 1-9). The *trps* have a characteristic helix-turn-helix xenobiotic response element (HTH-XRE) domain that is present in a subset of DNA-binding proteins. However, the *B. henselae trp* genes were poorly transcribed and did not seem to have a discernible native promoter region [76]. Xenobiotic response elements (XREs) are a family of transcriptional regulators shown to regulate biogenesis of type IV pili, flagella, and biofilm formation in other gram-negative bacteria [97]. This unusual arrangement of a stem-loop in the 3′ termini of the Brt RNAs and the apparent absence of a separate promoter for the DNA-binding protein *trp* genes prompted speculation that the Brt RNA may be a riboswitch or an RNA thermometer regulating the downstream *trp* genes and that the *brt-trp* tandem gene pairs may be involved in regulating *B. henselae* cellular processes. Riboswitches and RNA thermometers are regulatory cis-encoded sRNAs that fold into intricate structures, typically a stem and loop hairpin, in response to metabolites or environmental changes (pH, temperature) to modify the expression of a downstream gene [98,99]. A deletion of just the 3′ region of the *brt1* gene that forms the stem loop shows a high transcription rate of *trp1*, confirming that in the absence of the 3′ end of the RNA, the downstream *trp1* becomes highly transcribed [19]. Thus, the 3′ end of Brt1 acts as a transcriptional attenuator. Additionally, the resulting colonies from the 3′ deletion strain demonstrated increased aggregation and biofilm formation, suggesting a role in biofilm formation. Much like *B. henselae*, *B. bacilliformis*, the causative agent of Carrion’s disease, is transmitted to the human host by the sand fly arthropod vector [100]. Recently published data suggested that sRNAs in *B. bacilliformis* help with the adaptation of bacteria in different environmental conditions, necessary for survival in the vector and host [96]. This is achieved by differential expression of the sRNA depending on environmental conditions, a phenomenon also observed in *Borrelia burgdorferi* [101]. A comparison of the different *B. henselae* temperature and pH conditions mimicking both vector and host to determine the condition(s) that may enhance/eliminate the formation of the stem loop in our laboratory was not productive. Temperature ranges between 27 °C (arthropod vector) and 37 °C (mammalian host), and pH values between 6.6 and 7.2 to coincide with vector and human blood pH, respectively, did not show any significant differences in *trp* transcription or biofilm formation [16,101,102,103,104,105,106].

In a different study, an over-expressing strain of *trp1* showed increased biofilm formation, establishing that much like Brt1, the Trp1 protein is also involved in biofilm formation. Trp1 is a transcription factor (TF) annotated as a HTH, and most TFs are identified by the presence of a DNA-binding domain using sequence searches against protein family databases like NCBI-BLAST [100] and PFam [107]. HTH-XRE proteins have been shown to serve as a regulatory factor allowing adaptation between bacteria motility, adhesion, and biofilm formation [97,108,109]. Studies to identify the function of Trp1 showed that Trp1 binds to the promoter region of the *badA* gene, a TAA required for adhesion of the bacterial cell to the host cell and extracellular matrix proteins including fibronectin, for aggregation, and for inducing a pro-angiogenic host response [17,110,111].

*B. henselae* has two major virulence factors that play a role in *Bartonella* pathogenesis, VirB/T4SS [112] and BadA (Bartonella adhesin A). VirB/T4SS is a bacterial type IV secretion system (T4SS) that translocates DNA and protein substrates to host cells and requires cell-to-cell contact [113]. The VirB/T4SS system in *B. henselae* mediates invasion, proinflammatory activation, and anti-apoptotic protection of endothelial cells [112]. The regulation of *badA* is linked to the BatR/S two-component system, the general stress response system, and the Bartonella regulatory transcript [76,114,115]. The expression level of *badA* has been shown to correlate with robust biofilm formation [16,76,116]. Our group previously confirmed that *badA* is required for optimum biofilm formation [16]. An in-frame deletion of *badA* failed to form a stable biofilm, and a partial complementation of the gene partially restored biofilm formation. Also, the expression level of both *trp1* and *badA* were elevated in cells that formed biofilms in comparison to planktonic cells. Bacterial cells that lack the *badA* gene also failed to firmly adhere to a surface to form a stable biofilm. Other TAAs have been shown to influence biofilm formation in a variety of gram-negative bacteria such as *Salmonella enterica* [117], *Acinetobacter baumanii* [75], and *Burkholderia* species [118]. We also discovered that *badA* expression is significantly downregulated in the lab, under environmental conditions (low temperature and pH) that favor the arthropod lifecycle [16,101,102].

BadA is required for biofilm formation in *B. henselae*, and biofilms are a significant virulence mechanism for bacteria [16]. Biofilms are characteristically a stable community of bacterial cells involved in chronic infection and disease relapse. Biofilm cells exhibit increased resistance to the host immune response and antibiotic treatment. Microbial biofilms are associated with a range of disease conditions such as dental plaque, infections on medical devices, pneumonia, and infective endocarditis [20]. Biofilms are implicated in both the vector and host of *B. henselae* as well as persisting in the flea fecal matter, which is the inoculum passed onto cats and humans [39,119,120]. In humans, *B. henselae* is known to cause a range of diseases and/or symptoms (cat scratch disease, bacillary angiomatosis with neovascularization) [28,121,122,123,124]), and most notably, persistent bacteremia and infective endocarditis, two disease conditions that require *B. henselae* growth as a biofilm [125,126,127,128,129].

Investigation of biofilm formation in fleas and flea fecal matter shows that *B. henselae* can survive in both flea and flea feces up to 10 days post inoculum. Cat fleas were fed with cat blood infected with *B. henselae*. Florescent images show that the bacterial load persists in the flea gut for up to 10 days, and scanning electron micrographs show that biofilm is present in the fecal matter [19]. Gene expression levels in infected cat blood, in the fleas after ingestion, and in flea feces show that the *brt1* genes are highly expressed in the blood and result in a low expression rate for both the *trp1* and *badA* genes. Brt1 was only detected in the fleas 3 days after infection, with low expression profiles for *trp1* and *badA* genes, suggesting that the genes are not responsible for biofilm formation in the flea. In the fecal matter, *brt1*, *trp1*, and *badA* maintain the same expression profile for 2 days (high *brt1* and low *trp1/badA*), but from day 3, the fecal matter expression profile changes to decreased *brt1* and increased *trp1/badA*, a timeline that coincides with the formation of biofilms in the fecal matter [19]. An obvious difference in environmental conditions for blood/cat flea vs. feces is the presence of heme, a metabolite with concentrations that can go up to toxic levels—about 5mM in the arthropod vector after a blood meal [130,131]. *B. henselae* expresses hemin-binding protein C to protect from the toxic effect of the heme level [132]. An infection model for *Yersinia pestis*-infected fleas to study disease transmission to a host suggests that bacterial biofilm aggregates cause a blockage in the flea foregut leading to a regurgitation—a process by which the bacterial aggregate is transferred into the host during a blood meal on the host [133]. In contrast, flea feces are made up of hydrolyzed and partly digested blood, suggesting that heme toxicity is reduced in cat flea feces [134]; moreover, heme levels in humans are heavily sequestered and found at lower levels of about 0.5µM in blood [130,131], suggesting both conditions present non-toxic heme levels. *trp1* transcription was not increased under pH and temperature conditions that represent both arthropod- and mammal-like conditions, but data showed that as heme concentration decreased, a *gfp* gene cloned downstream of *brt*1 fluoresced intensely, backing up cat flea data that shows no *trp1* transcription in the presence of high heme [19]. As most riboswitches respond to metabolites, Brt1 responded to heme concentration, a condition that differs between the cat flea and mammalian hosts. This suggests that Brt1 as a *trp1* transcript attenuator may respond to heme concentration.

In addition to RNAs, which act as transcript terminators, most sRNA regulation occurs by base-pairing their target mRNA, prohibiting the target from being translated [135]. A few examples of sRNAs that can act as repressors and/or prompt translation of another mRNA were reviewed by Medha et al. [136] and Caron et al. [137]. In gram-negative bacteria, SgrS found in *E*. *coli* and *Salmonella* is the only well-characterized dual-function sRNA, where it represses or promotes the translation of four different mRNAs [136,138]. Another dual sRNA regulator, Agr, decreases the expression of many proteins on the cell surface and increases the expression of several virulence factors secreted, playing a central role in the pathogenesis of *S. aureus* [139,140].

Preliminary and unpublished RNA-seq data from our lab using biotinylated Brt1 RNA incubated with lysates of *Bartonella* biofilm cells and controlled with planktonic-grown cell lysates show that Brt1 can interact with mRNAs coding for surface adhesins and transporters including BadA mRNA, DNA binding proteins that included Trp7, heat shock chaperones, and mRNAs involved in metabolic function. Brt1 and BadA are located in different areas of the genome. This suggests that Brt1 may also regulate other distal genes as a trans-acting RNA, which base-pairs mRNAs to promote or suppress translation. Focusing on the relationship with the *trps*, especially with the tentative involvement of Brt1 with Trp7 mRNA, we observed that a *brt1* deletion did not affect *trp1* or *badA* transcription or biofilm negatively, instead, an increase in *trp1* expression and biofilm formation was observed confirming our earlier publication citing Brt1 as a negative regulator of *badA* [76]. As previously mentioned, *trp* genes are not abundantly expressed, but an over-expressing strain of the *trp1* gene shows that in a biofilm cell, *trp1* and *trp3* were abundantly expressed (Figure 2). Trp3 can bind upstream of the *badA* gene, as we confirmed by electrophoretic mobility shift assay and proteomics, indicating that at least two of the multicopy *brt/trp* genes may possess similar functions regulating *badA* expression.

RT-qPCR data using an overexpressing *trp1* strain (OE *trp*1, *Bh*/pNS2P_Trc_
*trp1*) constructed by cloning *trp1* upstream of the Trc promoter shows that both *trp1* (*p* = 0.000014), *trp3* (*p* = 0.008293), and *trp6* (*p* = 0.001020) expression is significantly upregulated in a biofilm cell (Figure 2). Trps 1 and 3 have already been implicated in regulating *badA* transcription. Moving forward, these preliminary Brt RNA-seq target data could be confirmed by RNA electrophoretic mobility shift assay (REMSA). We propose incubating Brt1 mRNA with BadA and Trp7 mRNA implicated by the RNA seq data to confirm interaction(s) identified by the RNA-seq. The result of this experiment may confirm whether Brt1 acts as a trans-acting mRNA by regulating the BadA mRNA, while providing an insight into the interaction between other Brts and Trps mRNA.

Hence, Brt1, Trp1, and BadA are all involved in biofilm formation and play a role in mammalian infection. The Brt1 RNA under conditions that favor the arthropod vector is highly expressed and forms a stem-loop that prevents downstream transcription of the *trp1* gene. Once conditions that favor mammalian hosts are met, namely low heme, high temperature, and neutral pH, the Brt1 loop yields for *trp1* transcription. Trp1 in turn binds the promoter region of the *badA* gene to facilitate adhesion, aggregation, and biofilm formation. This description was the first to implicate an sRNA, a DNA binding protein/transcription factor, and an adhesin in biofilm formation in *B. henselae*. This RNA and TFs may be present in multicopy to compensate for the loss of function. Unpublished RNA-seq data from our lab suggest that Brt1 can also bind Trp7, BadA, and other auto-transporters; however, these findings have not been confirmed. Other research has continued to study sRNAs that may influence bacterial survival under different environmental conditions.

## 5. Conclusions

The prevalence of flea-borne diseases such as CSD is consistently underestimated by health agencies [141,142]. Arthropod-borne viral or bacterial diseases are a constant source of public health concern and result in about 700,000 recorded deaths per year [143,144]. Advances in molecular genetics show that bacterial genomes may harbor multiple copies of their genes to help them adapt and survive in new niches–host–switching, signifying that there are regulatory circuits that recognize and coordinate expression of genes necessary for transition [95,145,146,147]. Biofilms are implicated in the infection process both in the vector and the host [19,148]. Biofilms are also considered the default mode of growth for bacteria [149], recognized as an essential survival mechanism for most bacterial infections [150]. Therefore, the regulatory mechanism of biofilm formation warrants study to better understand chronic bacterial infections, persistent bacteremia, and antibiotic resistance and tolerance.

In this review, we presented the clinical relevance of *B. henselae* biofilms, the different conditions required for biofilm formation, and the regulatory mechanism of the *brt1/trp1/badA* genes, which contribute to biofilm formation leading to optimal survival and fitness in the host environment. We discuss the absence of the *brt/trp/badA* in regulating the biofilm formation in the *C*. *felis* arthropod vector, indicating that perhaps a different outer membrane protein may serve as the adhesin for biofilm formation. We also discussed other genes and sRNAs implicated in regulating biofilm, *badA*, and/or involved in host infection and survival.

## Figures and Tables

**Figure 1 microorganisms-09-00835-f001:**
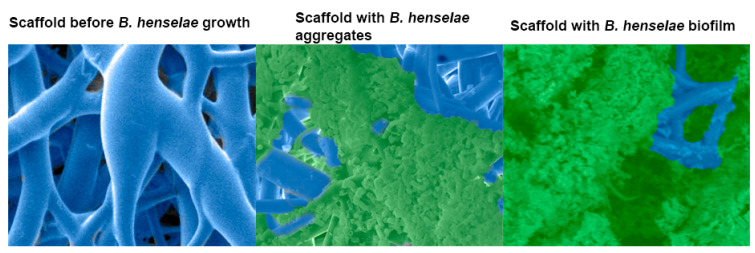
Scanning electron micrograph of a 48 h *Bartonella henselae* (*B. henselae*) biofilm growing on a 3-dimensional nanofibrous scaffold. Left, scaffold before bacterial growth. Middle, bacterial growth, adhesion, and aggregation around the scaffold branches. Right, *B. henselae* biofilm covering the scaffold and eclipsing the bacterial cells. Biofilm was preserved by the addition of the cationic dye, Alcian blue.

**Figure 2 microorganisms-09-00835-f002:**
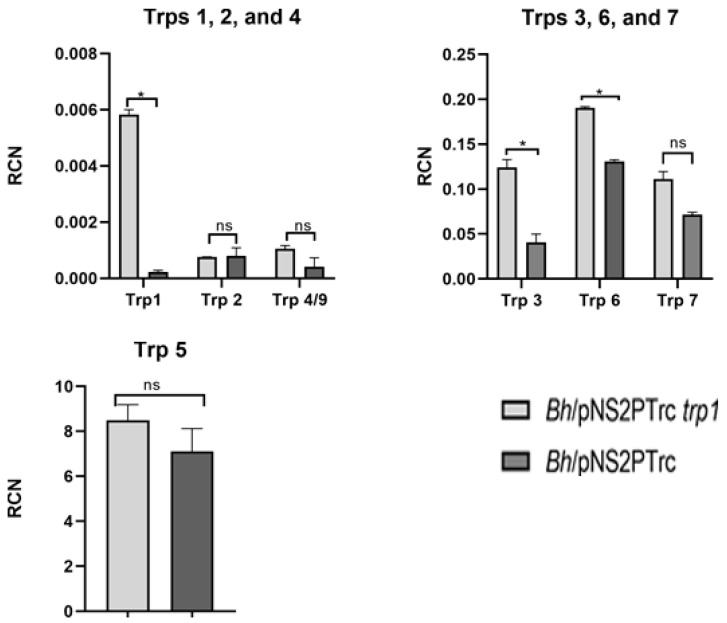
RT-qPCR showing expression of *trps* in *B. henselae* Houston-1/pNS2P_Trc_
*trp1* (overexpressing *trp1*) with *B. henselae* Houston-1 pNS2P_Trc_ as control. **Upper left**: Expression of *trps* 1, 2, and 4/9. **Upper right**: Expression of *trps* 3, 6, and 7. **Lower**: Expression of *trp* 5. Relative copy number (RCN) was compared to reference rplD mRNA. Bars represent means of three independent experiments, and error bars represent standard errors. Statistical analysis using Student’s *t*-test was performed using GraphPad Prism (GraphPad Software, San Diego, CA, USA), (*) with a *p*-value < 0.05 considered statistically significant as indicated.

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
