# Peer review of "What Is in a Cat Scratch? Growth of Bartonella henselae in a Biofilm"

_microorganisms, 2021, doi:10.3390/microorganisms9040835_

Round 1

Reviewer 1 Report

The review in the file below

Author Response

Response to reviewer 1.

Thank you for your helpful comments.   We have attempted to address your suggestions for improvement as follows:

  1. The Genus and species names have been italicized throughout the manuscript.
    1. Lines 14, 16 and 19 of the Abstract
    2. Genus and species names were not italicized in the references since all publication titles are already italicized.
  2. We have justified the legend for figure 1.  We have replaced the font above Figure 1 to correspond to the font in the text.  The file does not allow us to replace the actual figure in the manuscript but we have also submitted the revised figure for replacement by the editors.
  3. We have included additional references about the possible role of ticks in the transmission of henselae as requested by reviewer 2 as well as the reference you suggest (Angelakis et al).
    1. Reference 40, line 109: Zajac et al
    2. Reference 41, line 109: Cotte et al
    3. Reference 42, line 109: Angelakis et al
  4. We have expanded the text in the Introduction about disease presentations in cats and included the suggested reference.
    1. Expanded text on line 52-56
    2. Reference 13, line 56: Stitzer, B. and K. Hartmann

Reviewer 2 Report

The paper is well written, clear and targeted to a specific important aspect of CSD, so the review is useful and not repetitive compared to previous ones.

I would suggest to better clarify the title, helping the reader to immediately grasp the main subject of the study, as described at lines 75-77.

Some minor comments:

  • line 46 - "almost 50 defined or proposed species" - please cite some recent references where these 50 species are listed
  • line 97 - I suggest to change the sentence as follows: "Ctenocephalides felis (C. felis), known as cat fleas...".
  • line 98 - "other vectors as ticks have been proposed": I would like to see some more recent literature at this proposal. I can cite here two suggestiones: a correlation between human exposure to tick bites has been correlated to CSD (Zajac et al, 2015, J Vector Ecol) and transmission of Bartonella henselae in different life-stages of Ixodes ricinus has been demonstrated (Cotté et al, 2008, Emerg Infec Dis).
  • line 36, 38, 64, 109, 120, 136, 143, 149, 150, 165, 176, 185, 195, 217, 226, 275, 284, 357, 364, 366: spacing typing errors.

Author Response

Response to reviewer 2

Thank you for your helpful comments.   We have added a subtitle to more accurately describe the focus of the review (line 2).

Minor comments:

  1. We have included a recent citation describing the currently described Bartonella
    1. Reference 12, line 52: Okaro U et al
  2. This sentence has been changed as suggested.
    1. Line 107-108
  3. We have included the suggested citations describing the possible role of ticks in transmission of henselae.
    1. Reference 40, line 109: Zajac et al
    2. Reference 41, line 109: Cotte et al
    3. Reference 42, line 109: Angelakis et al
  4. The spacing errors have been corrected.
    1. Refer to track changes throughout